# Windows of opportunity for predicting seasonal climate extremes highlighted by the Pakistan floods of 2022

Nick Dunstone [1] ✉, Doug M. Smith [1], Steven C. Hardiman[1], Paul Davies[1], Sarah Ineson[1], Shipra Jain[2], Chris Kent[1], Gill Martin [1] & Adam A. Scaife [1,3]

Skilful predictions of near-term climate extremes are key to a resilient society. However, standard methods of analysing seasonal forecasts are not optimised to identify the rarer and most impactful extremes. For example, standard tercile probability maps, used in real-time regional climate outlooks, failed to convey the extreme magnitude of summer 2022 Pakistan rainfall that was, in fact, widely predicted by seasonal forecasts. Here we argue that, in this case, a strong summer La Niña provided a window of opportunity to issue a much more confident forecast for extreme rainfall than average skill estimates would suggest. We explore ways of building forecast confidence via a physical understanding of dynamical mechanisms, perturbation experiments to isolate extreme drivers, and simple empirical relationships. We highlight the need for more detailed routine monitoring of forecasts, with improved tools, to identify regional climate extremes and hence utilise windows of opportunity to issue trustworthy and actionable early warnings.

Seasonal climate prediction using dynamical global climate models is a well-established operational activity[1] coordinated by the World Meteorological Organisation (WMO). There are 14 global producing centres around the world (https://www.wmolc.org/) providing global seasonal forecasts every month. These forecasts are used by national meteorological and hydrological services and by WMO-organised Regional Climate Outlook Forums (RCOFs) to provide advanced warning of impending regional climate variability. However, seasonal climate variability is strongly flow-dependent, with more extreme regional climate anomalies occurring when the large-scale atmospheric circulation is perturbed. Seasonal prediction relies primarily on slowly evolving coupled ocean-atmosphere modes of variability, particularly the El Niño Southern Oscillation (ENSO), which can be skilfully predicted many months ahead[2]. The warm (El Niño) and cold (La Niña) phases of ENSO, which peak in amplitude during boreal winter, shift the location of regions of strong atmospheric convection and hence latent heating in the tropical Pacific, resulting in global changes in atmospheric circulation[3] and associated surface climate extremes[4]. Other coupled ocean-atmosphere modes, such as those in

the Indian and Atlantic oceans, can also drive remote teleconnections to regional climate variability in a similar manner[5].

ENSO events do not occur every year but typically every 2–7 years, so it is challenging to evaluate the conditional skill of seasonal predictions given the typical 20- to 30-year period of retrospective forecasts (hindcasts) used for assessment. Commonly used estimates of average forecast skill and reliability, calculated over all years, may therefore be overly pessimistic for years with active climate drivers such as ENSO events[6,7]. The standard way of communicating seasonal forecast information is via probabilities of different quantiles with respect to the climatological distribution. Terciles are the most common quantiles chosen. Whilst these terciles provide a level of discrimination that is typically well matched to the levels of average seasonal forecast skill, there exist 'windows of opportunity' where more confident warnings of more extreme climate events can be provided. Here we explore the extreme seasonal rainfall that led to widespread flooding over Pakistan in the summer 2022 monsoon season as an example of such a case.

[1]Met Office Hadley Centre, Exeter, United Kingdom. [2]Centre for Climate Research Singapore (CCRS), Singapore, Singapore. [3]University of Exeter, Exeter, United Kingdom. ✉e-mail: nick.dunstone@metoffice.gov.uk

Pakistan sits on the western edge of the South Asian monsoon system and typically has an arid summer climate. However, Pakistan can also be subject to extreme rainfall, as was the case in summers 2010[8,9] and 2022. Many studies have linked the 2010 extreme rainfall and circulation anomalies to a strong summer La Niña in the tropical Pacific and the resulting westward shifted West Pacific Subtropical High[10] as well as the wider ENSO teleconnection to south Asia, which is well represented in seasonal forecast systems[11]. In addition, an extratropical influence via upper troposphere circulation anomalies was also identified and linked the Pakistan floods to the summer 2010 Russian heatwave[12–14]. The 2010 event has been described as a 'freak incident'[15]. Yet only 12 years later, during the strongest summer La Niña since 2010, another even more devastating Pakistan flooding event occurred, causing a major humanitarian disaster: current estimates suggest over 1730 people died, 2.1 million were left homeless and with flood damages and economic losses estimated to be US$30 billion

(https://www.worldbank.org/en/news/press-release/2022/10/28/pakistan-flood-damages-and-economic-losses-over-usd-30-billion-and-reconstruction-needs-over-usd-16-billion-new-assessme). Here we focus on the extent to which seasonal forecasts were able to predict this extreme rainfall and lessons we can learn in order to issue more confident warnings of climate extremes in future.

## Results

### Observed and real-time operational forecasts of Pakistan summer 2022 rainfall

The 2022 Pakistan floods were unprecedented. The observed total rainfall during the summer (June–August, JJA) monsoon season was 415 mm which is 260% of, and 4.9 standard deviations (σ) above (Fig. 1b), the climatological mean rainfall. This was considerably more extreme than the impactful wet summer of 2010, which was a 2.4σ extreme event.

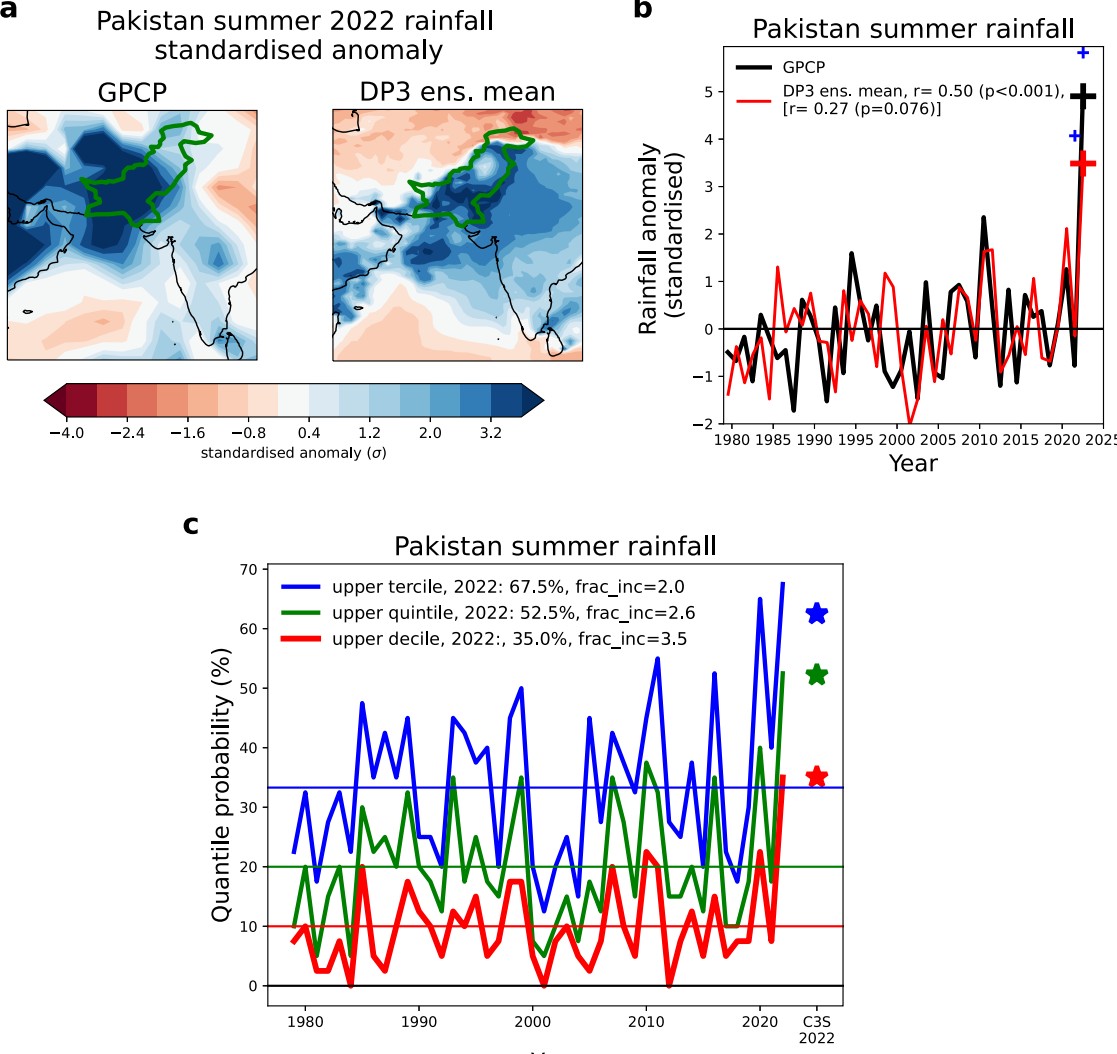

**Fig. 1 | Extreme Pakistan rainfall in summer 2022. a** Standardised rainfall anomaly maps for summer (June–August) 2022 for GPCP observations and the DP3 ensemble mean forecast. **b** Timeseries of observed and predicted standardised Pakistan summer rainfall anomalies with correlation coefficients shown both including and excluding summer 2022. Large crosses show 2022 values, and the small blue crosses in 2021 and 2022 show estimates of the impact due to La Niña from perturbation experiments, as discussed in the text. **c** Timeseries of probabilities of exceeding the upper threshold of different quantiles (upper tercile, quintile and decile) relative to the 1990–2020 DP3 climatology (horizontal lines). Star symbols show the corresponding multi-system mean quantile probabilities for summer 2022 Pakistan rainfall from the eight operational seasonal prediction systems in the C3S database (see "Methods") relative to their common 1993–2016 climatological period.

Seasonal forecast outlooks for Pakistan and the wider South Asian region did provide some warning of enhanced risk of rainfall ahead of summer 2022. The Pakistan Meteorological Department issued an outlook for summer 2022 (dated May 31, 2022, https://www.pmd.gov.pk/en/assets/seasonal-outlooks/Seasonal-June-July-August-2022.pdf) in which it forecast *"a tendency for above-normal precipitation is predicted over most parts of the country"* and that *"flash flooding in hill torrents of Punjab, AJK [Azad Jammu and Kashmir] and KP [Khyber-Pakhtunkhwa], also urban flooding in major plain areas of Punjab, Sind and KP cannot be ruled out."*. Similarly, the South Asian Regional Climate Outlook Forum (SARCOF, https://meteorology.gov.mv/downloads/89/view), in their pre-summer meeting (held during April 26–28, 2022), showed regional rainfall tercile probability maps with a >60% probability in the upper tercile leading them to forecast that: *"Normal to above-normal rainfall is most likely during 2022 southwest monsoon season over most parts of South Asia"*. The WMO Global Seasonal Climate Update (https://www.wmolc.org/gscuBoard/list), issued on May 26, 2022, stated that *"Parts of the Indian subcontinent and the southern Arabian Peninsula do have enhanced probabilities for above-normal rainfall but model consistency is only moderate."*.

The focus on tercile probabilities essentially gave a two in three chance for Pakistan summer rainfall to lie in the upper third of climatological Pakistan rainfall distribution. Given that the upper tercile is expected to occur on average one year in three, this forecast for a doubling of upper tercile probability may not have appeared overly concerning to many users and indeed little preparatory action was taken. We show below that signals for *extreme* rainfall, well beyond the upper tercile, were widely present in seasonal prediction systems in summer 2022 and suggest approaches that may enable more confident warnings to be issued in future.

## Looking beyond tercile probabilities

In this study, we focus on predictions made by the Met Office DePreSys3 near-term prediction system[16] (hereafter referred to as 'DP3'), initialised on 1st May, which has a 1-month lead-time ahead of summer. We use this system as it has a long 43-year (1979–2021) hindcast period, a large 40-member ensemble size and has been used previously for perturbation experiments to probe forecast signals and drivers[17,18], but similar forecasts were made by other systems (discussed below). Ensemble mean summer 2022 rainfall anomalies, in units of standard deviations of the ensemble mean hindcast variability (Fig. 1a), show extreme rainfall (+2-3σ) anomalies widely over Pakistan. The area-averaged ensemble mean rainfall over Pakistan (Fig. 1b) shows a +3.5σ anomaly for summer 2022, which, in agreement with the observed timeseries, is an unprecedented extreme over the last 44 years.

The DP3 upper tercile probability forecast for summer 2022 Pakistan rainfall of 67.5% (corresponding to twice the 33% climatological probability) is in good agreement with the operational multi-model seasonal forecasts discussed above (and shown in Fig. 1c as star symbols). However, to fully assess how extreme this forecast is, we need to probe further into the forecast probability distribution. An obvious question, one that a user may ask when presented with the raised summer 2022 upper tercile probability, is how does this forecast compare to previous years? The summer 2022 Pakistan rainfall upper tercile prediction of 67.5% is the highest probability over all 44 years considered (Fig. 1c, blue line)—eclipsing even the wet summers of 2010 and 2020. This additional comparison with previous years, not readily available from the real-time seasonal outlooks discussed above, is another clear indication that the summer 2022 forecast is unusually extreme.

We further examine even more extreme quantiles. The probability of the upper quintile Pakistan rainfall is 52.5% (Fig. 1c, green line), which is 2.6 times the 20% climatological probability, whereas the upper tercile probability was doubled. An even more extreme quantile, seldom used in seasonal forecasting, is that of deciles where we find an exceptional (35%) probability of the upper decile (Fig. 1c, red line), which is 3.5 times the 10% climatological probability. Note also that the 2022 forecast becomes more extreme relative to other years as higher quantiles are considered, e.g., the upper decile probability was only two times higher than climatology in 2010 and 2020. We note that these raised probabilities for the higher quantiles in 2022 are consistent with the >3σ anomaly in the DP3 ensemble mean (Fig. 1b).

Significantly raised probabilities for extreme quantile Pakistan rainfall in summer 2022 were widespread among the operational seasonal prediction systems and DP3 is very representative of the eight systems in the Copernicus Climate Change Service (C3S) archive (star symbols in Fig. 1c, see "Methods"). Remarkably, four of the nine systems considered here predicted summer 2022 to have an unprecedented high probability for upper decile Pakistan rainfall, with three of these systems having 50–80% of their ensemble members occupying the upper decile, which corresponds to five or more times the climatological values.

In summary, these results present a much more extreme view of Pakistan rainfall forecasts in summer 2022 compared to the tercile probabilities presented in the real-time forecast outlooks discussed above. Such extreme forecasts are potentially much more useful for motivating actions to reduce the impacts, but the key question is how much confidence do we have in them?

## Using physical understanding to build forecast confidence

The most common way of assessing forecast confidence is to assess the average skill over the hindcast period. The correlation between observed and forecast ensemble mean Pakistan rainfall is significant and reasonably high (r = 0.50, *p* < 0.001) over the 44-year hindcast (Fig. 1b). However, when 2022 is excluded to represent the situation prior to this summer's extreme, then the correlation is much lower (r = 0.27, *p* = 0.08). We note similar levels of skill are seen in the hindcasts of the eight C3S seasonal forecast systems (r = −0.02 to 0.38). Importantly, this average skill could severely underestimate the skill when strong climate drivers such as ENSO, or the Indian Ocean Dipole (IOD), are active and perturb the large-scale atmospheric dynamics in a predictable way. For example, the forecast ensemble mean correctly predicted increased rainfall in both 2010 and 2020 (Fig. 1b), two of the wettest Pakistan summers prior to 2022, which also coincided with strong summer La Niña events. If we had a sufficiently long sample of observed and forecast events, then we could quantitatively assess this state-dependent skill−i.e., how skilful Pakistan rainfall seasonal forecasts are when certain climate drivers (such as ENSO) are active. Unfortunately, the typical 20- to 30-year length of seasonal hindcasts gives too few cases to allow state-dependent skill to be robustly assessed. This assessment becomes even harder if there are two or more drivers that can combine to drive regional climate extremes. We suggest a possible way forward is to build a collection of evidence to assess forecast confidence based on a physical understanding of the dynamical mechanisms.

The forecast low-level (850 hPa) circulation anomalies in summer 2022 are very similar (pattern correlation, r = 0.7) to that of the ERA5[19] observational reanalysis (Fig. 2a, b). Anomalous easterly flow is seen extending from the Bay of Bengal across northern India towards Pakistan, similar to that seen in summer 2010[10], creating anomalous convergence of moisture over Pakistan. In addition, the anomalous easterlies promote the northwest track of monsoon depressions (low-pressure systems) from the Bay of Bengal into the monsoon trough and towards northern Pakistan[20], as was observed in both summer 2010 and 2022. The anomalous easterlies stretch further east to the Philippines, with an anomalous south-westerly flow also seen over China. Together these anomalies are indicative of an intensified and westward-shifted West Pacific Subtropical High (WNPSH). Furthermore, anomalous south-westerly flow is seen over the Arabian Sea, corresponding to an increase in the strength of the Somali jet (or 'low-

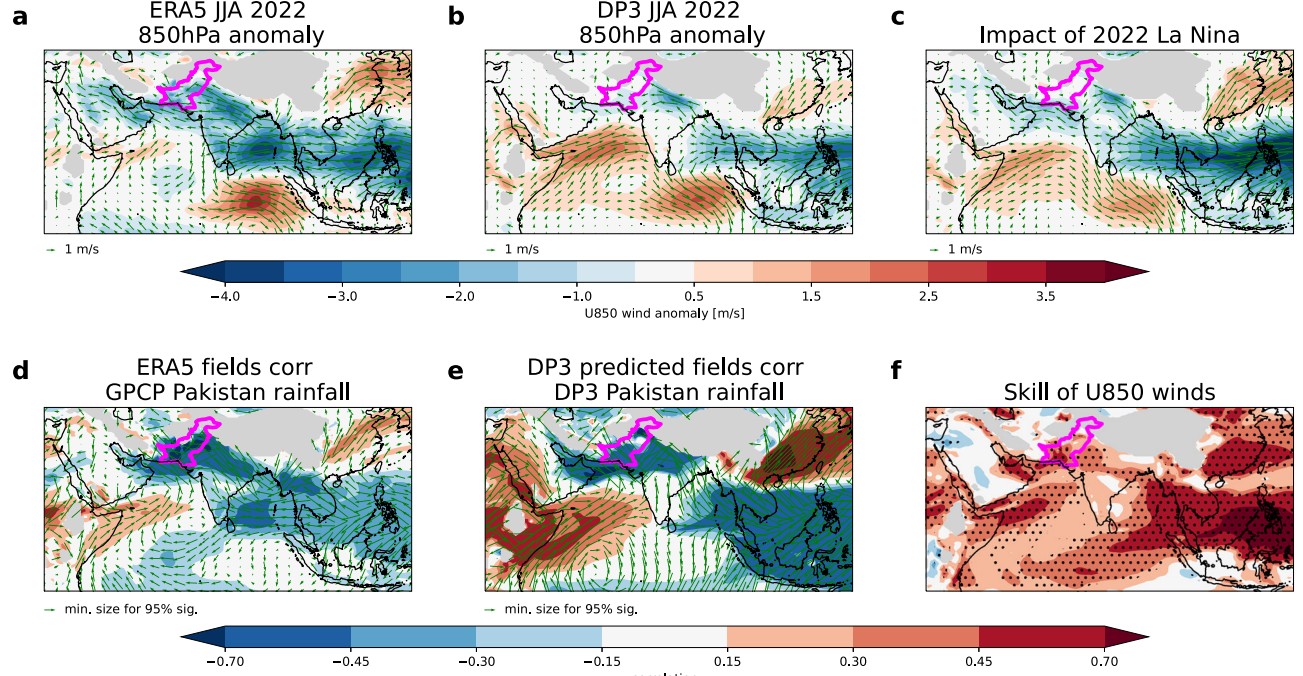

**Fig. 2 | Low-level circulation associated with Pakistan summer rainfall.**
**a**, **b** Maps of 850 hPa wind anomalies in summer 2022 for ERA5 (**a**) and DP3 ensemble mean forecast (**b**) with zonal winds (shading) and wind vectors (arrows) and Pakistan outlined in magenta. **c** As (**a**, **b**) but for an estimate of the impact of La Niña conditions from perturbation experiments (as discussed in the text). **d**, **e** Historical (1979–2022) correlation between Pakistan summer rainfall timeseries and the field of 850 hPa zonal winds (shading) and wind vectors (arrows) in ERA5 (**d**) and the DP3 ensemble mean (**e**). **f** Gridpoint correlation skill for DP3 ensemble mean in predicting ERA5 observed 850 hPa zonal winds. Stippling shows correlations significantly different from zero at the 95% confidence level according to a two-sided Student's *t*-test.

level' jet), which increases the advection of moisture towards north-west India and Pakistan[21].

Crucially, all of the key low-level circulation features identified in 2022 are also seen in the historical teleconnection between Pakistan rainfall and circulation in observations and DP3 predictions (Fig. 2d, e) and have been identified previously, showing that the forecast for 2022 was consistent with known drivers. Furthermore, they exhibit significant average skill over the hindcast period (Fig. 2f).

We similarly analyse the upper-level (250 hPa) circulation anomalies (Fig. 3). The observed and predicted summer 2022 anomalies both show a poleward shift of the subtropical Asian jet, giving anomalous easterlies to the south and westerlies to the north. The observed anomalous easterlies were significantly stronger than those predicted by DP3 and reversed the climatological westerly flow over the southern Tibetan Plateau. This reversal in upper-level winds has been linked to anomalous ascent and hence extreme rainfall over Pakistan in summer 2022[22]. Again, the circulation anomalies in 2022 compare well to the historical teleconnection between Pakistan rainfall and upper-level circulation (Fig. 3d, e). The DP3 predictions of the zonal wind anomalies are also skilfully predicted (Fig. 3f), particularly on the southern flank of the subtropical jet.

To better understand the connection between the upper and low-level circulation anomalies, we assess the latitude-height cross-section over Pakistan (Fig. 4). The observed and predicted 2022 anomalies are again very similar, with a barotropic easterly flow connecting the low and upper levels and anomalous ascent (arrows) over Pakistan. The poleward shift of the subtropical jet is also clearly visible in the upper troposphere. The 2022 anomalies are very similar to the historical teleconnection of the field with Pakistan rainfall in both observations and DP3 prediction and zonal winds are skilfully predicted in the key regions (Fig. 4d–f).

Overall, the excellent agreement between the predicted 2022 atmospheric circulation anomalies and the historical teleconnections

and skill maps over the Indo-Pacific region shows that the forecast was consistent with known drivers and helps to build confidence in it. We further note that this evidence is not particularly sensitive to the inclusion of 2022 (Supplementary Fig. S1). However, teleconnection patterns do not necessarily establish causality, and we assess additional experiments to build further confidence in the drivers in the next section.

**Perturbation experiments to further assess mechanisms**
As mentioned earlier, 2022 saw the strongest summer La Niña since 2010 in both the observations and DP3 ensemble mean predictions. Forecasts of summer ENSO variability are highly skilful (Niño3.4 region skill: r = 0.86, p < 0.001), giving us significant confidence in the La Niña prediction. Given the existing literature connecting La Niña with Pakistan flooding[10] and the evidence above (Figs. 2–4), further perturbation experiments are warranted to verify its role in driving the 2022 forecast. Similar perturbation experiments have been performed in previous retrospective case studies to isolate and understand the impact of different drivers, for example, the influence of a sudden stratospheric warming on the European winter of 2005/6[23], the strong North Atlantic sea surface temperature (SST) tripole effect on European summer 2018[17] and the influence of the extreme IOD event on European winter 2019/20[18]. We suggest that such perturbation experiments to establish physical drivers and mechanisms of responses to large-scale SST anomalies could, in theory, be performed in real time in order to be able to issue more confident warnings of impending extreme events.

To assess the influence of the summer 2022 La Niña, we nudge the ocean temperature and salinity in the tropical Pacific towards a recent near-neutral ENSO year (taken as 2021 here). Nudging is performed to create initial conditions on May 1, 2022, that do not contain the strong signal of summer 2022 La Niña, while the rest of the ocean, sea-ice and the atmospheric initialisation remain unchanged from the original forecast (see "Methods" for further details). We then run a 40-member ensemble parallel to the original forecast and assess the impact of La

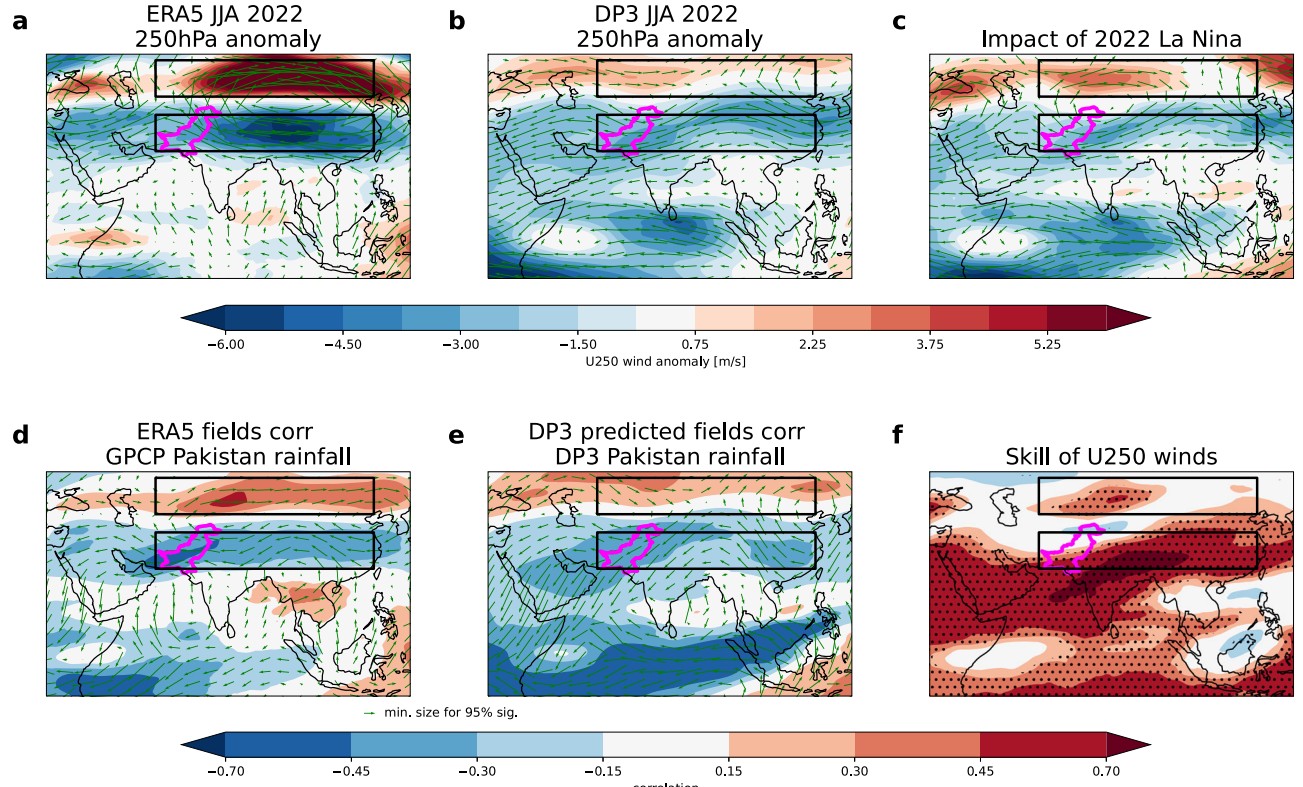

**Fig. 3 | Upper-level circulation associated with Pakistan summer rainfall.**
**a**, **b** Maps of 250 hPa wind anomalies in summer 2022 for ERA5 (**a**) and DP3 ensemble mean (**b**) with zonal winds (shading) and wind vectors (arrows) and Pakistan outlined in magenta. **c** As (**a**, **b**) but for an estimate of the impact of La Niña conditions from perturbation experiments (as discussed in the text). **d**, **e** Historical (1979–2022) correlation between Pakistan summer rainfall timeseries and the field of 250 hPa circulation in ERA5 (**d**) and the DP3 ensemble mean (**e**). **f** Gridpoint correlation skill for DP3 ensemble mean in predicting ERA5 observed 250 hPa zonal winds, stippling shows correlations significantly different from zero at the 95% confidence level according to a two-sided Student's *t*-test. Magenta boxes on all panels show the boxes used to define the subtropical jet meridional shift index (STJshift, see "Methods").

Niña as the original forecast minus this experiment. It is possible that any perturbation may simply destroy the predictable signals such that the difference does not represent the influence of La Niña. To address this, we repeated the experiment in reverse by nudging the 2022 tropical Pacific initial conditions into May 1, 2021, initial state and so diagnosed the influence of La Niña again as the difference between this experiment and the original 2021 forecast.

The estimated impact of La Niña on DP3 Pakistan rainfall from both perturbation experiments confirms its strong influence on the 2022 forecast (shown by the two small blue '+' signs in Fig. 1b). The impacts of La Niña on the 2022 large-scale atmospheric circulation (Figs. 2–4 panel c) are also consistent with the actual forecast (Figs. 2–4 panel b), and historical teleconnections[21] (Figs. 2–4 panels d and e). This attribution of the forecast signals to the predicted summer La Niña further strengthens our physical understanding. If these experiments had been run in real-time, ahead of the summer, they would have increased our confidence that the unusually strong forecast summer La Niña was a significant driver of the predicted extreme Pakistan rainfall signals.

**Using a simple empirical model to gain understanding**
Simple empirical models (e.g., using multiple linear regression) can be useful tools to assess the impact of multiple drivers on year-to-year variability. Above, we identified both the strength of the WNPSH and a meridional shift in the subtropical Asian jet as two large-scale circulations driving Pakistan summer seasonal rainfall. We create a WNPSH index[24] and a subtropical jet meridional shift (STJshift) index using the boxes shown in Fig. 3 (see "Methods"). As expected, these two indices are both significantly correlated with observed Pakistan rainfall

(Fig. 5c, r = 0.34 and r = 0.51 for the WNPSH and STJshift, respectively). They are largely independent (cross-correlation = 0.15) and provide a multiple linear regression model with high correlation (r = 0.67). We note that the correlation drops to r = 0.53 (*p* < 0.001) when 2022 is excluded but remains highly significant. Furthermore, the influence of the two drivers becomes more similar (r = 0.31 and r = 0.34 for the WNPSH and STJshift, respectively) when 2022 is excluded. Although simple, this empirical model does help us to understand some of the past variability of Pakistan rainfall. For example, when WNPSH and the STJshift acted in opposite directions, such as in the strong La Niña of summer 1998, extreme rainfall did not occur.

The WNPSH is highly predictable (Fig. 5a, r = 0.72, *p* < 0.001) over the hindcast period, consistent with previous studies examining the WNPSH as a driver of summer Chinese climate[25]. The skill in predicting the STJshift index is weaker but still highly significant (Fig. 5b, r = 0.48, *p* = 0.001). We note that if we exclude 2022, we find no drop in skill for the WNPSH and a relatively small drop for the STJshift index (r = 0.41, *p* = 0.007).

Using the multiple linear regression model with the DP3 ensemble mean predicted 2022 standardised anomalies for the WNPSH and STJshift indices gives a greater than two standard deviation forecast for Pakistan rainfall in 2022 (Fig. 5c magenta cross). Note that both indices are in phase in summer 2022, both acting to increase Pakistan rainfall, thereby providing further evidence to support the extreme prediction from the dynamical model.

## Discussion
We have shown that strong signals for *extreme* rainfall that led to the summer 2022 Pakistan floods were present in current seasonal

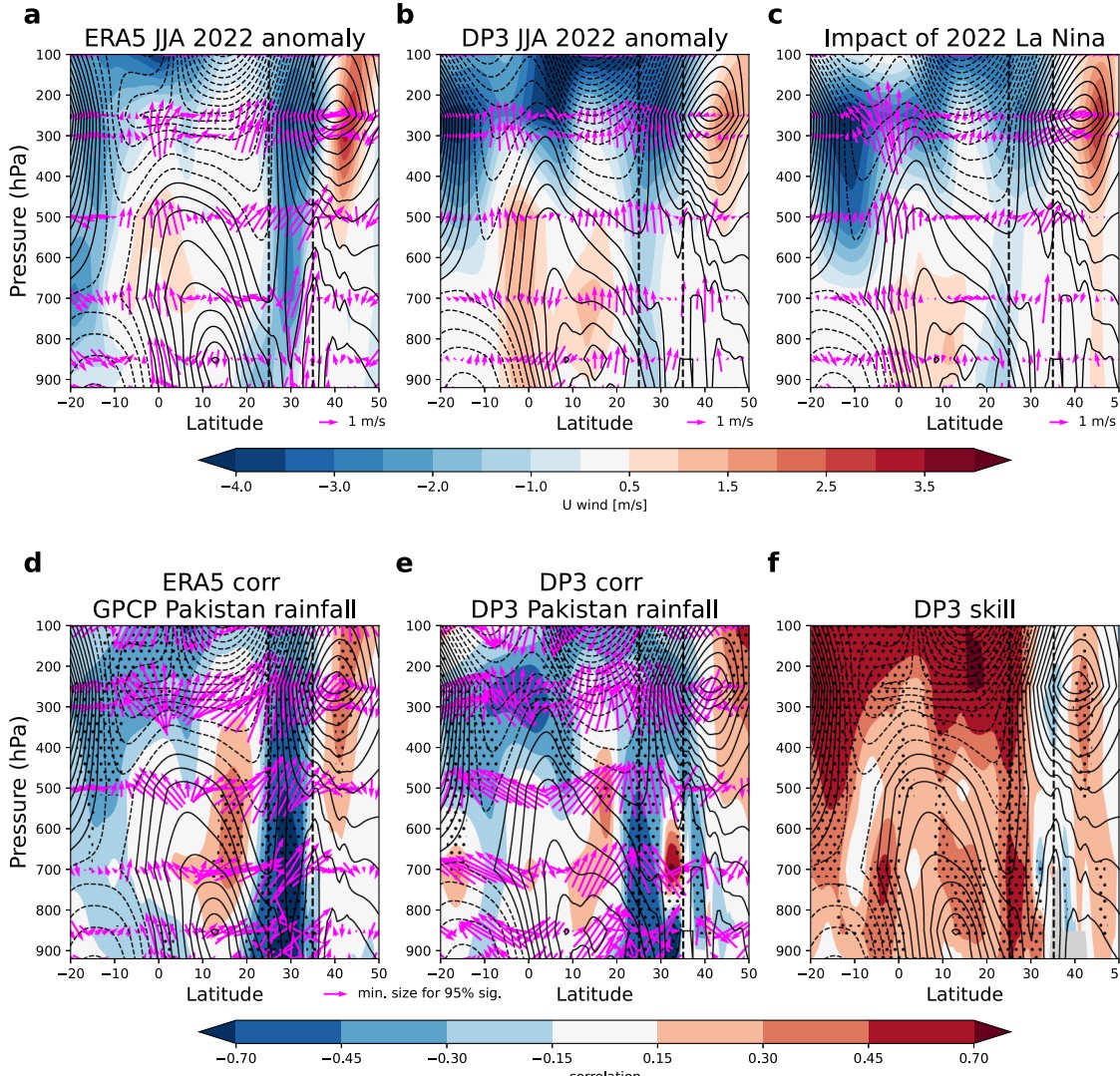

**Fig. 4 | Cross-sections of circulation associated with Pakistan summer rainfall.**
**a**, **b** Latitude-height cross-sections over Pakistan (longitudes 60-75E) of summer 2022 wind anomalies for ERA5 (**a**) and DP3 ensemble mean (**b**) with zonal winds (shading) and wind vectors (arrows). **c** As (**a**, **b**) but for an estimate of the impact of La Niña conditions from perturbation experiments (as discussed in the text). **d**, **e** Historical (1979–2022) correlation between Pakistan summer rainfall time-series and circulation in ERA5 (**d**) and the DP3 ensemble mean (**e**). **f** Gridpoint correlation skill for DP3 ensemble mean in predicting ERA5 zonal winds, stippling shows correlations significantly different from zero at the 95% confidence level according to a two-sided Student's *t*-test. Contours in all panels show the climatological zonal winds, with solid lines showing positive (westerly) winds and dashed lines showing negative (easterly) winds. Dashed vertical lines in all panels show the approximate latitudinal extent of Pakistan.

prediction systems. Whilst the real-time operational seasonal outlooks provided good advice for an enhanced risk of above-normal (or upper tercile) Pakistan rainfall, they did not explicitly signal a high risk of *extreme* rainfall. Had a fuller examination of the forecast distribution been considered, for example, using higher quantiles, then the greatly increased risk of extreme rainfall, with multiple systems showing unprecedented upper decile probabilities, could have been identified. We therefore suggest that near-term climate forecasts should be routinely monitored for extreme signals and discuss ways to build confidence in such forecasts.

Climate extremes, by definition, are rare events, and so assessing skill is very challenging from a sample size of 20–30 cases (years) typically provided by seasonal hindcasts. This is particularly true when an extreme arises due to the combination of drivers acting together. However, by examining the physical drivers of the Pakistan rainfall, we suggest that summer 2022 was a window of opportunity for increased forecast confidence. There was a particularly strong summer La Niña, which our perturbation

experiments show promoted a combination of a strong WNPSH and a poleward shifted subtropical Asian jet that provided the large-scale dynamical conditions leading to the extreme Pakistan rainfall. However, whilst La Niña may have been necessary to drive these circulation features in 2022, it is not likely a sufficient condition alone, and other perturbation experiments could be performed to understand the influence of other drivers such as those from the North Atlantic region[26] and the extent to which they were simulated. The regional large-scale circulation drivers are consistent with historical Pakistan rainfall variability and are skilfully predicted by DP3, and therefore can be used to provide alternate lines of evidence to build confidence in the extreme forecasts.

Recent work has shown other windows of opportunity when predictable factors (e.g., the IOD and North Atlantic SSTs) drive changes in atmospheric circulation[17,18,27]. Here and in these earlier studies, perturbation experiments can be used to attribute regional climate extremes to remote drivers to understand the physical mechanisms driving predictable climate variability. We suggest that

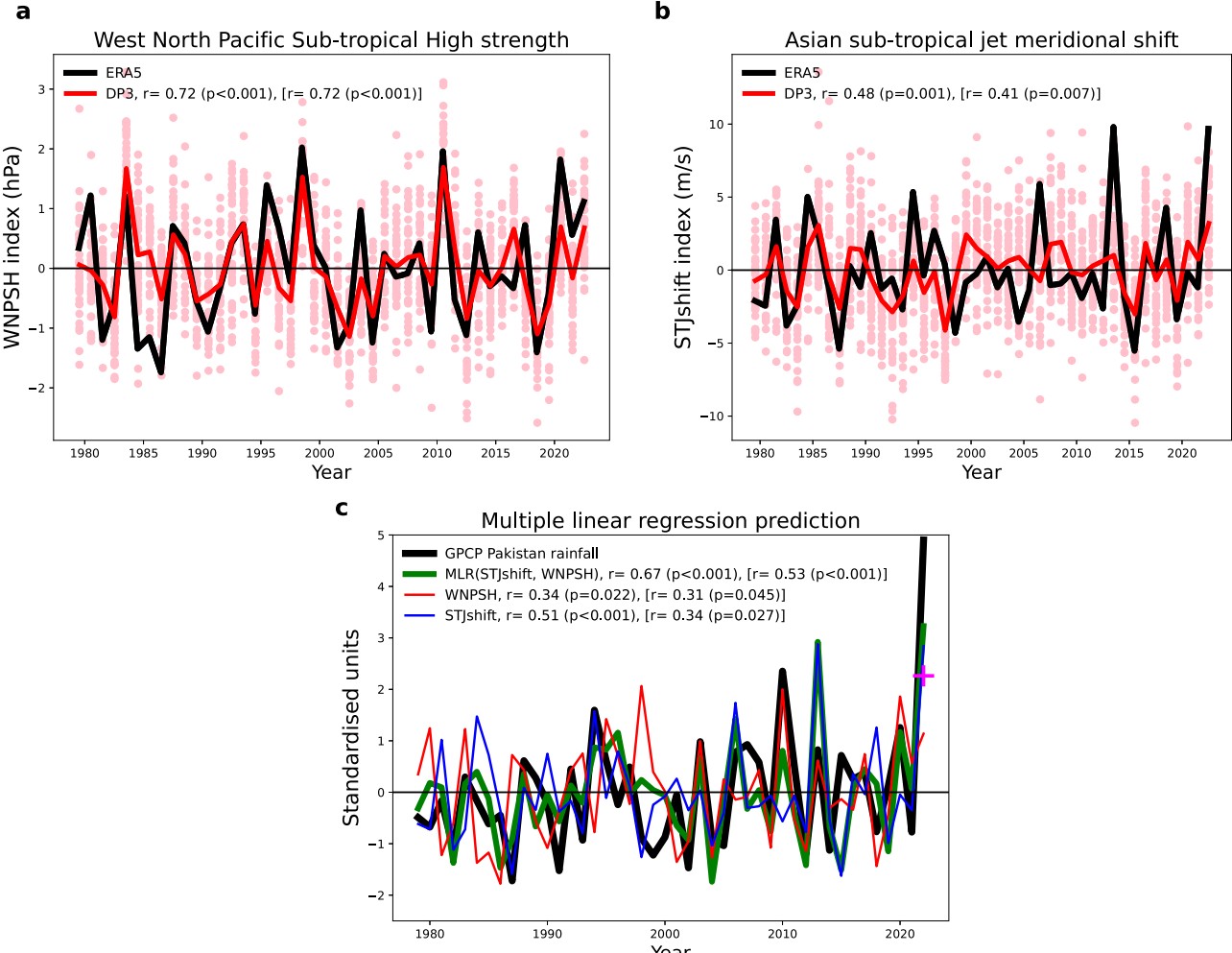

**Fig. 5 | Simple empirical model for understanding forecast signals.**
**a** Standardised timeseries of the west north Pacific subtropical high strength (WNPSH) in ERA5 (black) and DP3 forecast ensemble mean (red) with individual ensemble members shown by pink circles. **b** As (**a**), but for the Asian subtropical jet meridional shift (STJshift) index. **c** Standardised timeseries of observed Pakistan summer rainfall (black), west north Pacific subtropical high (red), STJshift (blue) and the resulting multiple linear regression timeseries (green) using these two indices to predict Pakistan rainfall. The magenta cross in 2022 indicates the value obtained from using the 2022 ensemble mean predicted values of the two indices as input to the multiple linear regression model.

such experiments could even be performed in real-time in order to build physical confidence in forecast extreme signals.

It is, of course, easy to be wise after the event with the benefit of hindsight, but it is important to learn lessons so that windows of opportunity can be fully utilised to issue more confident early warnings in future. Such a task is potentially very labour-intensive but could be helped by new tools that routinely identify extreme forecast signals and also provide a preliminary analysis of physical drivers to build confidence.

Here we present an example of such a tool, where the first step is to identify extreme forecast signals by plotting global maps of standardised ensemble mean forecast anomalies aggregated over climate regions (as shown in Fig. 6 using the DP3 summer 2022 forecast). For this example, we use previously defined regions which divide the global land into 237 regions of approximately equal area (0.5 Mm²) which were specifically designed for the analysis of climate extremes[28] (see "Methods"). Each region is approximately the size of a medium-to-large country, for example, Pakistan is itself a single region. An advantage of these regions being based on political/economic boundaries is that they naturally align with geographical domains of decision-making, e.g., for disaster preparedness. The standardised forecast rainfall anomalies for summer 2022 show that Pakistan is one of only four regions with a >

±3σ signal, whilst a further 31 regions have >±2σ rainfall anomalies (Fig. 6). The number of extreme regions identified is much larger when predictable drivers such as ENSO are active, as in summer 2022. Whereas many summers have very few regions with significant predicted extremes—for example, in 2013, only one region had a >±2σ rainfall anomaly (Supplementary Fig. S2).

We propose an interactive version of Fig. 6 where each region can be clicked to provide further automated analysis for that region. This key second analysis step is aimed at probing the full forecast probability distribution and driving physical mechanisms. Data could include timeseries such as that in Fig. 1b, c, allowing the user to view forecast signals from a historical perspective, view changes in quantile probabilities, assess average hindcast skill and whether previously observed extreme years were well predicted. Other information that could be displayed includes historical correlation maps between local rainfall and the large-scale atmospheric circulation (as shown in Figs. 2 and 3, panels d–f) and relationships to known drivers such as ENSO, the IOD and SST patterns. This could then be augmented by a third analysis step of perturbation experiments, when appropriate, to attribute forecast signals. Such a tool would therefore help with the efficient identification of forecast climate extreme signals, including the filtering of false alarms by exploring the physical drivers and so increase forecast confidence.

## Summer 2022 forecast rainfall anomalies

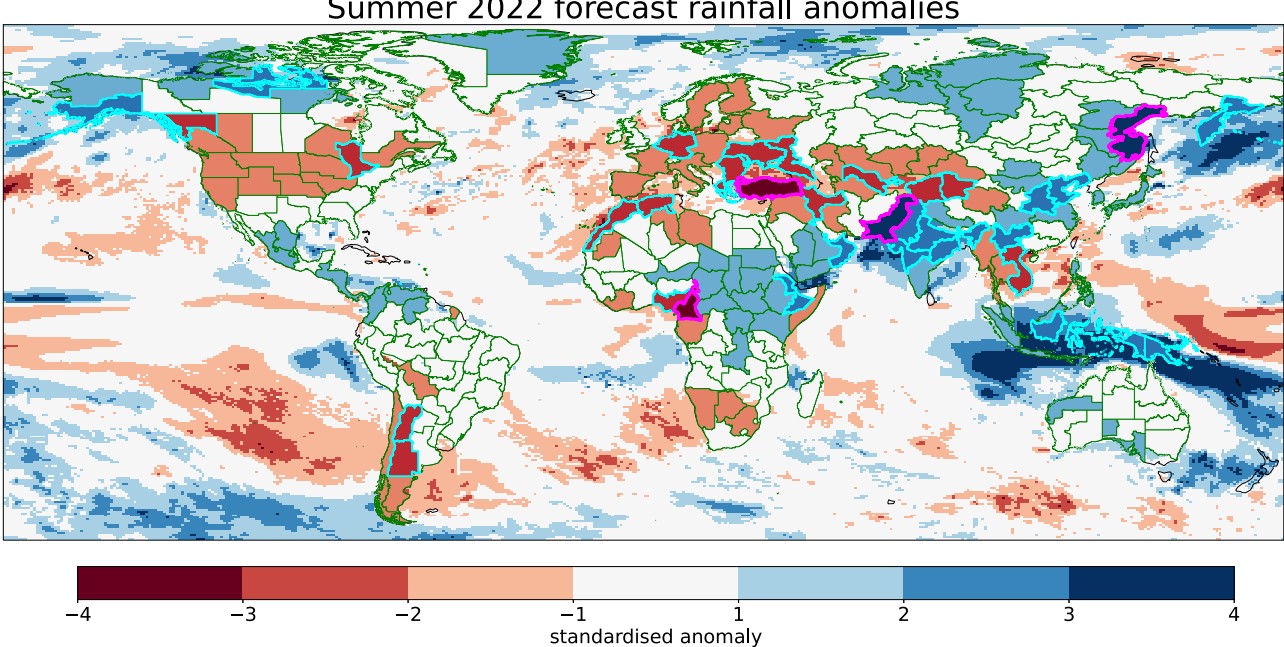

**Fig. 6 | An interactive tool to identify climate extremes.** DP3 ensemble mean standardised predicted rainfall anomalies for summer 2022 from 1st May initialised predictions are shown (see "Methods"). Global land is split into 237 regions[28] of approximately equal area (0.5 Mm²) with regional averages shown and green lines dividing regions. The 31 regions with a >±2σ anomaly are highlighted by a thicker cyan border and the four regions with a >±3σ are highlighted with a magenta border. On an interactive web version of this map, the user can 'click' on any region to open a dedicated webpage showing further information aimed at assessing forecast confidence (see text), and a mask layer can be applied to only show those regions with statistically significant skill over the hindcast period (Supplementary Fig. S3).

The true value of such a tool would, of course, need to be tested and verified in real-time forecasts. However, we assess the potential utility of the tool's first step by evaluating the observed quantile outcomes (using GPCC dataset[29] over the 1979–2022 period) when extreme (>±2σ) wet or dry summers are identified in the DP3 hindcast ensemble mean (see "Methods"). As an example, we apply a modest minimum average correlation skill threshold of r > 0.25 to filter for regions that show some evidence (90% confidence) of significant rainfall skill. This gives 77 regions (Supplementary Fig. S3), including Pakistan, with a total of 134 extreme rainfall events predicted over the 44 years. The resulting observed frequency of correct outer quantile outcomes (Fig. 7a) is many times greater than their climatological probabilities. The most frequent outcome is the outer quintile which happens in over half of events. In contrast, the probability of an incorrect outer quantile outcome is much reduced. The ratio of correct outcomes to climatological frequency increases as we examine more extreme quantiles, with an outer decile extreme occurring in more than a third of events and the outer 5th percentile occurring in more than a fifth. As the average correlation skill threshold is increased, the percentage of successful outer quantile outcomes also increases (Fig. 7b), though fewer regions/ extremes are forecast (Fig. 7b, magenta line).

In summer 2022 35 regions (Fig. 6) have a predicted >±2σ rainfall anomaly, of which 16 show evidence for significant skill in the hindcast period (r > 0.25, see Supplementary Fig. S3). Examining the observed GPCC summer 2022 outcomes, we find that 12 of the 16 (75%) regions experienced a correct sign outer quintile rainfall event, of which eight (50% of the 16) were outer decile and four (25% of the 16) were outer 5% rainfall extremes (including Pakistan). The frequency of these extreme observed quantile outcomes is considerably higher than climatological expectation, in agreement with the analysis in Fig. 7, and the four of 16 that did not experience an observed correct outer quintile event were near-neutral outcomes (fell into the middle three quintiles).

Our results demonstrate the potential utility of this tool and highlight that windows of opportunity exist for skilful predictions of some boreal summer rainfall *extremes*, including the extreme Pakistan rainfall in summer 2022. We note that the second step of the tool, the interactive assessment of the hindcast performance and physical mechanisms would, via expert judgement, help to further filter false alarms and increase forecast confidence.

Here we have focussed on ways to extract the *maximum information* from our seasonal predictions, particularly for climate extremes. However, even the most skilful seasonal forecasts will only help society if they are acted upon by those that receive and use them. Whilst beyond the scope of this study, we highlight the increasingly important role of social science in developing an understanding of how best to communicate seasonal forecast information and build user confidence. We acknowledge that forecast 'false alarms' are a key concern and must be carefully assessed in developing and issuing early warnings of climate extremes. Nevertheless, given the potential demonstrated here and in other studies, we strongly recommend further using windows of opportunity to build confidence in seasonal forecasts of climate extremes to issue trustworthy and actionable early warnings.

## Methods
### Data and indices
Observed rainfall data are taken from the Global Precipitation Climatology Project (GPCP) dataset[30], allowing the examination of rainfall over the wider Pakistan region, including over the ocean. In the evaluation of the extremes tool over land regions (Fig. 7), the Global Precipitation Climatology Centre (GPCC) dataset[29] is used. Observed winds and mean sea level pressure (MSLP) are taken from the ERA5 reanalysis[19].

The primary prediction system used here is the Met Office DePreSys3 system (DP3[16]) based on the HadGEM3-GC2 coupled climate model[31]. The atmosphere has a horizontal resolution of approximately 60 km (and 85 vertical levels) and an ocean resolution of 0.25° (75 vertical levels). A full-field data assimilating simulation is

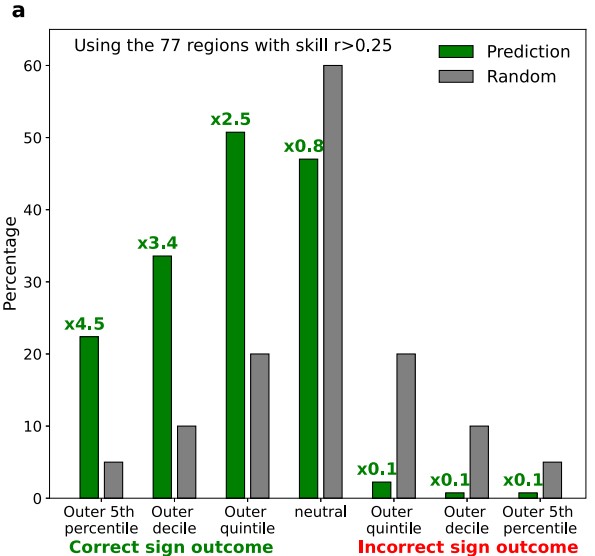

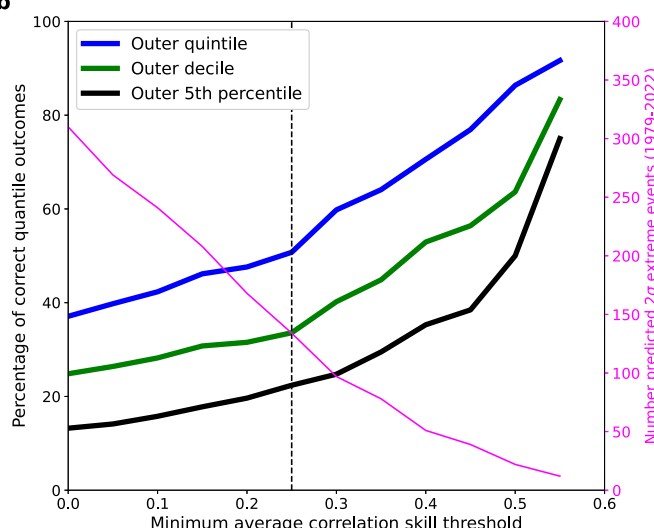

**Fig. 7 | Assessing the utility of the proposed extremes tool. a** Predicted seasonal extreme (>±2σ) DP3 ensemble mean rainfall signals are compared (over 1979–2022) with observed rainfall quantile outcomes for the 77 regions that have a statistically significant average correlation skill over the hindcast (r > 0.25, Supplementary Fig. S3) giving 134 predicted extreme events in total. Green bars show quantile outcome frequencies with the text above giving the ratio relative to the climatological quantile probabilities. The 'neutral' case refers to the central three quintiles. **b** shows how the percentage of correct quantile outcomes increases as a function of the minimum average correlation skill threshold. The magenta line (right axis) shows how the number of predicted extremes reduces as less regions are included in the analysis. The vertical dashed line highlights the r > 0.25 threshold illustrated in panel (**a**).

performed where the climate model is nudged in the ocean, atmosphere and sea-ice components towards observations. In the ocean, temperature and salinity are nudged towards a monthly analysis created using global covariances[32] with a 10-day relaxation timescale. In the atmosphere, temperature and zonal and meridional winds are nudged towards the ERA-Interim[33]/ERA5[19] reanalysis with a 6-h relaxation timescale. Sea ice concentration is nudged towards monthly values from HadISST[34] with a 1-day relaxation timescale. Hindcasts are started from the 1st May initial conditions of this assimilation simulation and a 40-member ensemble is created using randomly generated seeds to a stochastic physics scheme. The assimilation run, and hindcasts, have full knowledge of external forcing data sets (for example, greenhouse gases, aerosols, ozone, solar and volcanic forcings) as per the CMIP5 protocol[35] and follow the representative concentration pathway (RCP4.5) after 2005.

Eight additional operational seasonal prediction systems are assessed for their Pakistan rainfall forecast with data obtained via the Copernicus Climate Change Service (C3S) website. The 1st May 2022 forecast and corresponding hindcasts (1993–2016) are used from the ECMWF SEAS5 system[36], Météo-France System 8[37], the Centro Euro-Mediterraneo sui Cambiamenti Climatici SPS3.5[38], the Deutscher Wetterdienst GCFS2.1[39], the National Centers for Environmental Prediction CFS version 2[40], the Japan Meteorological Agency CPS3[41] and the UK Met Office Global Seasonal Forecast system version 6[42].

The strength of the WNPSH is calculated by averaging MSLP over the region [115-150E, 15-25N][24]. We create a subtropical jet meridional shift (STJshift) index by taking the difference in 250 hPa zonal wind averaged over the purple boxes shown in Fig. 3. The ENSO Nino3.4 index is defined over the region [170-120W, 5S-5N].

### La Niña perturbation experiments
To assess the impact of the summer 2022 La Niña on Pakistan rainfall and associated dynamical circulation anomalies, we performed perturbation experiments using DP3. These used a similar methodology to previous studies examining the impact of the North Atlantic SST tripole in summer 2018[17] and the IOD in winter 2019/20[18]. DP3[16] creates initial conditions for forecasts by nudging the climate model towards

an optimally interpolated ocean analysis[43] with a 10-day relaxation timescale. We modify the ocean analysis in the tropical Pacific, over a region 20S-20N with a 5° latitude ramp, by swapping the ocean analysis in this region with that of 2021 (a neutral ENSO summer) for the months starting in February and leading up to 1st May. The rest of the ocean, sea-ice and the atmospheric initialisation remained unchanged from the original forecast. After the initial conditions were created on 1st May, we ran a 40-member ensemble parallel to the original forecast. The impact of La Niña is assessed as the original forecast minus the experiment (i.e., with La Niña minus without La Niña). In addition, to assess the impact of the methodology, we performed the opposite experiment—nudging the 2022 tropical Pacific into the 2021 initial conditions. The impact of La Niña is then assessed as experiment minus original. In common with previous studies using this method, nudging only achieved approximately half of the desired signal (change in tropical SSTs). This is because the ocean relaxation is relatively weak (10 days), and no changes are made to the overlying atmosphere, which is still nudged as in the original forecasts. Therefore, to provide the best estimate of the impact of La Niña, and in common with previous studies[44], we inflate the signals according to the ratio of desired and actual Niño3.4 values obtained (approximately a factor of two).

### Extremes tool
The proposed interactive tool presented in Fig. 6 is an example of how extreme forecast rainfall could be identified and examined over regions corresponding to medium-sized countries or major provinces of larger countries. For this, we use the previously defined equal-area regions of Stone[28], but we note that the authors remain neutral with regard to jurisdictional claims in all maps. When examining both the regional extreme rainfall ensemble mean predictions (e.g., for summer 2022 in Fig. 6), and observed outcomes anomalies (including climatological observed quantile thresholds calculated over 1979–2021), we use the semi-standard deviation to separately calculate standardised anomalies on the wet and dry sides of distribution due to the skewed nature of rainfall in many regions. The use of the semi-standard deviation gives an approximately equal number of wet and dry

extremes. In Fig. 7, we evaluate the first part of the proposed tool by assessing how well the identified extreme DP3 predicted signals (>±2σ) correspond to extreme outcomes over the 44-year hindcast period. We note that land regions in Antarctica are excluded from the analysis (Figs. 6 and 7) as insufficient verifying observations are available. We use all other regions; however, we note that this could be filtered further in future by applying a minimum threshold on climatological rainfall totals, thereby excluding extremely dry regions.

## Data availability

ERA5 reanalysis data was downloaded from the European Centre for Medium-Range Weather Forecasts (ECMWF), Copernicus Climate Change Service (C3S) at Climate Data Store (CDS; https://cds.climate.copernicus.eu/). Data from the eight operational seasonal prediction systems are also available to download from the Copernicus data portal (https://climate.copernicus.eu/seasonal-forecasts). Global Precipitation Climatology Centre (GPCC) rainfall data over global land regions was downloaded from Deutscher Wetterdienst (DWD; https://www.dwd.de/EN/ourservices/gpcc/gpcc.html). Global Precipitation Climatology Project (GPCP) rainfall data was downloaded from the US National Oceanic and Atmospheric Administration (NOAA), National Centers for Environmental Information (NCEI, https://www.ncei.noaa.gov/products/global-precipitation-climatology-project). The shape-files used to define the global land regions are available in the supplementary data of Stone[28] (https://link.springer.com/article/10.1007/s10584-019-02479-6). Met Office DePreSys3 data used in this study are available online (https://doi.org/10.5281/zenodo.8380700).

## Code availability

The computer code used to produce the figures is available from the corresponding author upon request.

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

## Acknowledgements
This work was supported by the Met Office Hadley Centre Climate Programme funded by BEIS and Defra. It was also funded by the Met Office Climate Science for Service Partnership (CSSP) China project under the International Science Partnerships Fund (ISPF) and the Horizon Europe ASPECT Project (Grant No. 101081460).

## Author contributions
N.D. led the analysis. N.D. and D.S. wrote the first draft. S.H., P.S., S.I., S.J., G.M. and A.S. contributed to the editing and writing of the manuscript. C.K. provided technical assistance in creating the example extremes tool.

## Competing interests
The authors declare no competing interests.
