## [Peer Review File · Nature Communications]

REVIEWER COMMENTS

Reviewer #1 (Remarks to the Author):

Review comments of the manuscript NCOMMS-23-02675 entitled “Windows of opportunity for predicting seasonal climate extremes: Pakistan floods of 2022” by Dunstone et al. submitted to nature communications

The present study describes the extreme seasonal rainfall that happened in Pakistan during the summer of 2022 based on statistical tools and attempts to predict the climate extremes based on modelling simulations. They made the statistical assessment of different quantiles of seasonal rainfall during historical periods in addition to the real-time forecasts. The authors also tried to examine the physical drivers of the Pakistan rainfall through summer La Niña and a combination of a strong WNPSH and a poleward shifted sub-tropical jetstream, which provides the large-scale dynamical conditions leading to the extreme Pakistan rainfall. They propose real-time perturbation experiments to attribute regional climate extremes to different remote drivers to understand the physical mechanisms of seasonal variability. However, the present work has multiple shortcomings, which are given below, and those lead to the non-acceptance of the manuscript.

1. The manuscript is not having a clear and direct answer to the problem of how to give a prediction of seasonal extremes.
2. The way of writing the manuscript is not up to the mark of nature communication, and it is tough to follow the scientific content in the manuscript.
3. There was no mention of the days of Pakistan extreme events in the summer monsoon season of 2022. Since the manuscript primarily addresses the issue of these extreme events, which lead to seasonal extremes, and therefore, it is necessary to unravel the meteorological and oceanographic conditions of the extreme rainfall events.
4. In the examination of the Figure 1 a & b, the rainfall simulated by the ensemble model is not captured correctly. It is also better to give the bias of the model ensemble mean to assess the deviations of the model outputs from the observations.
5. In the analysis of multiple regression analysis, the authors used only two parameters (WNPSH and STJ shift) for the analysis; however, it is not mentioned how these parameters are helpful for an extreme seasonal rainfall prediction. How much prior to the event can it be predicted?
6. The second paragraph explained in the results section is not actually the findings of the authors, and hence that is not appropriate there.
7. Line 97: What are AJK and KP? not mentioned before

8. Lines 133-134: "...but similar forecasts were made by other systems". What is meant by "other systems"? Similarly, it was not mentioned in the manuscript or in the supplementary material about the simulation of individual members to create 40 member ensemble, only mentioned some references from which the methodology was adopted.
9. Line 135: what is meant by $+2-3\sigma$?
10. Lines 135-136: "The corresponding predicted time series of...". At least explain how the predicted time series is constructed.
11. The impact of La Nina was also given in Figure 2; however, a detailed explanation was not given, such as how it influenced the forecast of the forthcoming extreme event and how much weightage can be given to the intensity and duration of La Nina years for getting a reasonable prediction.
12. While doing the MLR analysis, this impact is not considered. Why?
13. How the model skill is estimated. It is better to give a brief note on it in the manuscript, either in the main or in the supplementary material.
14. Figures 3 and 4 give almost similar information. Then why is it necessary to keep both figures in the manuscript
15. The prediction based on the MLR with the WNPSH and STJshift indices is not exactly matching with the GPCP rainfall. It needs to find the cross-correlation between them to explore the strength of the relationship in the time domain.
16. Line 374-377: Not clear
17. The attempt to produce an interactive version of the spatial climate extremes is worth mentioning. However, the produced spatial maps are not exactly matching with the observed pattern, and therefore, the production of the interactive maps may be leading to wrong results. So it is better to try to reduce the model bias before going to interactive spatial maps.

Reviewer #2 (Remarks to the Author):

Review of "Windows of opportunity for predicting seasonal climate extremes : Pakistan floods of 2022" by Dunstone et al. submitted to Nature Communications (April 2023)

This paper investigates possible improvements of seasonal forecasts of tropical rainfall 1-4 months mostly from the case study of the extreme JJA season in 2022 in Pakistan. This season was somewhat well predicted but the amplitude of the extreme positive anomaly was underestimated. As seasonal forecasts use ensemble of simulations, the extremes are, by definition, underestimated as soon as the ensemble shows some spread, which is always observed. The authors propose to use supplementary experimental simulations (based on the modification of the initial atmospheric

state), when an extreme event is expected as in the boreal spring of 2022, to increase the confidence of the predictions of extremes. It may be an extremely helpful exercise for any impact study related to the occurrence of this extreme. Nevertheless, I do not find here a major and novel result demonstrating a practical improvement of seasonal forecasts. The main issue is that I do not really understand how the increased confidence related to the supplementary simulations is practically implemented. I mean that I do not see how the idealized experiments add something new to our knowledge about air-sea coupling (since we already know from myriad past diagnostic and numerical studies that the atmospheric initial state is not important when the SST forcing is large as it is expected during extreme ENSO events, so what ?) and, more important, how these idealized experiments are practically used to modify quantitatively the expected probabilistic forecasts ? I add that I do not see what you expect to be different between a seasonal forecast showing a large absolute anomaly and an idealized experiment as the one used in this study. I think it is a rather circular exercise. If the ensemble shows a large anomaly, it means a large (and robust) SST forcing upon the atmosphere, right ? The point of the practical implementation of the information gained from the idealized experiments, somehow a sort of post-processing the seasonal forecasts, is crucial to corroborate what is announced in the title of this ms. I add also that considering a single case study, even if it is the highest observed seasonal anomaly ever observed, is risky. For example, the figure 6 shows that strong positive anomalies are forecasted also over NE India and Bangladesh. Then, the 2022 monsoon was anomalously dry over this area. I am curious if the supplementary simulations show also positive forecasts here. If so, it casts some doubts about the real advantage (or the robustness) of the supplementary experiments to gain confidence in the seasonal forecasts, since in this case, the forecasts for the same season over a close region to Pakistan is totally wrong. It may be different if the authors are able to demonstrate the advantage of supplementary idealized experiments to get a better forecast and explain clearly how these better forecast are expressed in practical terms. They can choose to consider the same country with other extreme ENSO years or showing another close test area. So, by now, I found the idea and storyline interesting but I am not convinced by the results, especially how the idealized experiments are practically used to alter the probabilistic forecasts from the seasonal exercise. Another main concern is that the authors consider three possible drivers of the anomalous seasonal rainfall in Pakistan (La Nina event, zonal shift of the west Pacific anticyclone and meridional shift of the subtropical jet) without any clear description and explanation of their relationships, except the fact that both atmospheric indices are independent to each others. Are these atmospheric indices related to the ENSO phenomenon ? SO in summary, I think that this paper needs at least a major revision.

Figure: seasonal rainfall anomaly in JJA 2022 in mm (see [https://iridl.ldeo.columbia.edu/SOURCES/.NOAA/.NCEP/.CPC/.CAM5_OPI/.v0208/.anomaly_9120/.prcp/T/\(days since 1960-01-01\)streamgridunitconvert/T/differential_mul/T/\(months since 1960-01-01\)streamgridunitconvert/T/3/1.0/runningAverage/3/mul//units/\(mm/season\)def//long_name/\(Precipitation Anomaly\)def/DATA/-1000/-900/-800/-700/-600/-500/-400/-300/-200/-100/-50/50/100/200/300/400/500/600/700/800/900/1000/VALUES/prcp_anomaly_max1000_colors2/a/-a/+X+Y+fig+colors+|+thin+grey+contours+black+thin+solid+coasts+countries+fig+//aproduct/1000/1000/plotrange/T/last/plotvalue/X/-20/340/plotrange/Y/-65/75/plotrange+//plotborder+72+psdef//antialias+true+psdef//plotaxislength+700+psdef//XOVY+null+psdef//color_smoothing+null+psdef+.png?T=Jun-Aug+2022&plotaxislength=960](https://iridl.ldeo.columbia.edu/SOURCES/.NOAA/.NCEP/.CPC/.CAM5_OPI/.v0208/.anomaly_9120/.prcp/T/(days since 1960-01-01)streamgridunitconvert/T/differential_mul/T/(months since 1960-01-01)streamgridunitconvert/T/3/1.0/runningAverage/3/mul//units/(mm/season)def//long_name/(Precipitation Anomaly)def/DATA/-1000/-900/-800/-700/-600/-500/-400/-300/-200/-100/-50/50/100/200/300/400/500/600/700/800/900/1000/VALUES/prcp_anomaly_max1000_colors2/a/-a/+X+Y+fig+colors+|+thin+grey+contours+black+thin+solid+coasts+countries+fig+//aproduct/1000/1000/plotrange/T/last/plotvalue/X/-20/340/plotrange/Y/-65/75/plotrange+//plotborder+72+psdef//antialias+true+psdef//plotaxislength+700+psdef//XOVY+null+psdef//color_smoothing+null+psdef+.png?T=Jun-Aug+2022&plotaxislength=960))

More specific concerns

- line 80-82: is there an available reference for that statement ?
- Fig. 1: perhaps, you can state precisely that "summer" is JJA. Even it is stated in the method section, the captions should be more precise about the parameters of the scheme (initialization date, size of the ensemble, period of the retrospective forecasts, etc.) since a figure with its caption should be understood alone.
- line 128: initialization is May 1st ?
- line 188: you stated before that you have a 44-yr ensemble for one model. It is enough to see what happens in other extreme La Nina events.
- Fig. 2d.e: color are for zonal component only ? The caption is not fully clear.
- line 202: from which test ?
- line 204: do you have quantitative estimates (RMSE ? Pattern correlations ?) of the „similarity“ ?
- Fig. 4: the magenta arrows are difficult to depict.
- line 274: which indicator for ENSO ? SST indices ? From which initialization ?
- line 311: are these atmospheric drivers related to ENSO ?
- line 317: This score should be cross-validated, but this is perhaps the case ?
- line 358 and 361-362: same comment as line 311.
- line 365: I do not understand how the gain in confidence is implemented in the seasonal forecasts ?
- line 384: I do not understand if this map has been made from the seasonal forecasts only ? From the SF + supplementary experiments ?

Reviewer #3 (Remarks to the Author):

In this paper, Dunstone et al. explore the extreme seasonal rainfall associated with the 2022 Pakistan floods as an example of how windows of opportunity can be used to issue more confident forecasts of extreme climate events. The Met Office DePreSys3 near-term prediction system is used to forecast Pakistan rainfall. Eight additional operational seasonal prediction systems are also assessed. Focusing on upper quintile and decile probabilities, the authors show that signals for

extreme rainfall were widely present in seasonal prediction systems in summer 2022 and could have been identified had a fuller examination (beyond the upper tercile) of the forecast distribution been considered. Confidence in the DePreSys3 forecasts is established through an understanding of the physical drivers of the Pakistan rainfall, developed using perturbation experiments to isolate the effects of La Niña, and a multiple linear regression model relating rainfall to low- and upper-level circulation. Based on their results, the authors suggest that near-term climate forecasts should be routinely monitored for extreme signals and propose an interactive tool to aid with this.

The paper is well-written with the methods and results clearly presented. Given the extent of the damage that can be caused by extreme events (as discussed in the Introduction), issuing trustworthy and actionable warnings is of the utmost importance. There is great potential in the methods discussed here to develop confident forecasts of climate extremes. Thus, I believe this paper represents a valuable contribution to the field and will be of wide interest, meriting publication in Nature Communications.

I have only a small number of very minor comments/corrections:

The authors might consider referring to DePreSys3 using an abbreviation such as “DP3” (as in Ref. 18) instead of the “model” to distinguish it more clearly from the other seasonal prediction systems and the empirical model also used in the study.

Line 80: Correct “devasting” to “devastating.”

Line 146: Change “blue curve” to “blue line” for consistency with the next paragraph.

Line 153–4: “While we note that any shift will give a monotonically increasing factor on the likelihood to higher and higher percentiles [...]” I am not sure I fully understand what is meant by this. Could you please explain?

Line 281: Remove the citation in parentheses.

We thank the Editor and three Reviewers for their consideration of this manuscript. In addressing your comments, we believe the revised version of the paper is now more robust and provides stronger evidence beyond the example of the Pakistan summer 2022 extreme. To do this we have performed new analysis to evaluate the performance of the proposed interactive 'extremes tool' by examining the observed rainfall quantile outcomes when extreme regional rainfall signals are forecast over the 1979-2022 hindcast period for all regions. The results are shown in a new figure (Fig. 7) and highlight significant potential for identifying greatly increased risk of regional extreme rainfall quantiles (outer quintile, decile and 5th percentile) that could supplement, and go well beyond, the current standard analysis of terciles.

REVIEWER COMMENTS

Reviewer #1 (Remarks to the Author):

Review comments of the manuscript NCOMMS-23-02675 entitled "Windows of opportunity for predicting seasonal climate extremes: Pakistan floods of 2022" by Dunstone et al. submitted to nature communications

The present study describes the extreme seasonal rainfall that happened in Pakistan during the summer of 2022 based on statistical tools and attempts to predict the climate extremes based on modelling simulations. They made the statistical assessment of different quantiles of seasonal rainfall during historical periods in addition to the real-time forecasts. The authors also tried to examine the physical drivers of the Pakistan rainfall through summer La Niña and a combination of a strong WNPSH and a poleward shifted sub-tropical jetstream, which provides the large-scale dynamical conditions leading to the extreme Pakistan rainfall. They propose real-time perturbation experiments to attribute regional climate extremes to different remote drivers to understand the physical mechanisms of seasonal variability. However, the present work has multiple shortcomings, which are given below, and those lead to the non-acceptance of the manuscript.

We thank the reviewer for their detailed suggestions, we address these below.

1. The manuscript is not having a clear and direct answer to the problem of how to give a prediction of seasonal extremes.

Prediction of seasonal extremes is a very challenging problem and not one that a single paper can 'answer'. However, we believe that this paper highlights current shortcomings of the prediction of seasonal extremes and outlines some tangible steps forward. We believe that the new analysis (Fig. 7) provides more quantitative evidence, using all regions and the last 44 years, that skilful predictions of some seasonal extreme rainfall events are possible when large model predicted signals are present and that we should be taking better advantage of these windows of opportunity.

2. The way of writing the manuscript is not up to the mark of nature communication, and it is tough to follow the scientific content in the manuscript.

We have tried to clarify aspects in the revised version, although we note Reviewer 3's comment that "the paper is well-written with the methods and results clearly presented".

3. There was no mention of the days of Pakistan extreme events in the summer monsoon season of 2022. Since the manuscript primarily addresses the issue of these extreme events, which lead to

seasonal extremes, and therefore, it is necessary to unravel the meteorological and oceanographic conditions of the extreme rainfall events.

We know of several papers (submitted or under review) that discuss the detailed synoptic weather events that led to the extreme rainfall and flooding in Pakistan last year – we will include references if these are accepted/published. In particular, the record number of low-pressure systems that tracked across Pakistan last summer was the clear direct synoptic driver (we do touch on this in L214-216). However, from what we know of these studies, they all point to the large-scale seasonal circulation anomalies (the strong anomalous easterly circulation over India and the strengthened Somali jet that together resulted in enhanced moisture convergence over Pakistan) as the driver for this synoptic behaviour and so it is the seasonal mean rainfall and circulation patterns that are the focus of this paper.

4. In the examination of the Figure 1 a & b, the rainfall simulated by the ensemble model is not captured correctly. It is also better to give the bias of the model ensemble mean to assess the deviations of the model outputs from the observations.

Both the JJA 2022 seasonal prediction and observed outcome have large rainfall signals over Pakistan that are unprecedented since at least 1980. As expected for a seasonal forecast, the rainfall patterns do not match exactly, for example the prediction system gave signals for wetter conditions over North-East India and Bangladesh than observed. We now provide a more comprehensive evaluation of the performance over all regions/years in a new Fig. 7. We note that, as is standard practice in seasonal forecasting, we allow for model biases by assessing the magnitude of the respective signal in 2022 compared to the model climatological variability.

5. In the analysis of multiple regression analysis, the authors used only two parameters (WNPSH and STJ shift) for the analysis; however, it is not mentioned how these parameters are helpful for an extreme seasonal rainfall prediction. How much prior to the event can it be predicted?

The multiple regression analysis is not presented to give a forecast itself but rather as a tool to understand the dynamical model forecast Pakistan rainfall signal and hence give us improved physical confidence in the dynamical forecast. The red lines in Fig. 5a,b are the DePreSys3 ensemble mean prediction at a one month lead time (i.e. months 2-4, JJA) of the WNPSH and sub-tropical jet shift index and both are skilfully predicted ($r=0.72$ and $r=0.41$ respectively, both $p<0.01$). The predicted values in 2022 are used in the multiple linear regression model to give the magenta cross shown in Fig. 5c, demonstrating that they combine to explain a large fraction of the dynamical forecast signal. We have changed the labels on Fig. 5 for additional clarity.

6. The second paragraph explained in the results section is not actually the findings of the authors, and hence that is not appropriate there.

We think that it should be obvious to the reader that this is not our work, and it is important to set the scene of the operational seasonal outlooks in 2022. However it is not really background 'introduction' material – so we propose leaving it where it is for now but are happy to move it based on editorial recommendation.

7. Line 97: What are AJK and KP? not mentioned before

Good point, this is a direct quote from the source material but we have now expanded these in square brackets “[]” to aid the reader, thanks.

8. Lines 133-134: “...but similar forecasts were made by other systems”. What is meant by “other systems”? Similarly, it was not mentioned in the manuscript or in the supplementary material about the simulation of individual members to create 40 member ensemble, only mentioned some

references from which the methodology was adopted.

Slightly below this we discuss and show signals from a further eight operational seasonal systems in L164-171 and have now added “(discussed below)” to signal this on L136. The DePreSys3 system details are outlined in Dunstone et al 2016 as referenced, but we have now included further information in the Methods section about the system.

9. Line 135: what is meant by $+2-3\sigma$?

10. Lines 135-136: “The corresponding predicted time series of...”. At least explain how the predicted time series is constructed.

We’ve amended these lines to make this clearer.

11. The impact of La Nina was also given in Figure 2; however, a detailed explanation was not given, such as how it influenced the forecast of the forthcoming extreme event and how much weightage can be given to the intensity and duration of La Nina years for getting a reasonable prediction.

12. While doing the MLR analysis, this impact is not considered. Why?

The perturbation experiments, shown in Fig. 1b and panel (c) of Figs. 2-4, show that in summer 2022 the presence of the cold tropical Pacific SST anomalies (i.e. the La Nina) was required to generate the extreme model forecast Pakistan rainfall signals. However, that does not mean that the presence of the La Nina alone was a sufficient driver and other influences may also promote favourable circulation (particularly for the position/strength of the sub-tropical Asian jet) or they may oppose the La Nina influence in some other years. In terms of average correlation in the observations across the 44 years, then both the sub-tropical jet shift index and the WNPSH index are more strongly correlated with Pakistan rainfall ($r=0.51$ and $r=0.34$) than the ENSO Nino3.4 index itself ($r=-0.30$) and this is why they are used in the MLR analysis and not the ENSO directly. However, as shown in panel (c) of Figs. 2-4, at least in 2022, the La Nina was required to generate both the sub-tropical jet shift and WNPSH anomalies that year and hence the presence of a strong summer La Nina increased the probability of an extreme Pakistan rainfall event. Although we are very limited on space to expand on these points, we have added more details and references in L372-375.

13. How the model skill is estimated. It is better to give a brief note on it in the manuscript, either in the main or in the supplementary material.

We have added additional detail in Figs 2-4 figure captions to explain that it is the gridpoint correlation skill that is being assessed in these skill maps.

14. Figures 3 and 4 give almost similar information. Then why is it necessary to keep both figures in the manuscript

We think it is important to show both the spatial structure of the upper-level wind anomalies as a map in Fig. 3 and to show how these connect to the lower-level circulation using the cross-sections in Fig. 4.

15. The prediction based on the MLR with the WNPSH and STJshift indices is not exactly matching with the GPCP rainfall. It needs to find the cross-correlation between them to explore the strength of the relationship in the time domain.

The MLR model using WNPSH and STJshift indices have a very significant correlation of $r=0.67$ with Pakistan rainfall but this means that they explain just under 50% of the observed variance – so not an exact match. The cross-correlation between the WNPSH and STJshift indices ($r=0.15$) was already quoted in the text (now L323), showing that these two indices provide significant independent information.

16. Line 374-377: Not clear

We have now removed this sentence.

17. The attempt to produce an interactive version of the spatial climate extremes is worth mentioning. However, the produced spatial maps are not exactly matching with the observed pattern, and therefore, the production of the interactive maps may be leading to wrong results. So it is better to try to reduce the model bias before going to interactive spatial maps.

We have undertaken new analysis to evaluate the performance of this proposed new tool across regions and the 44 year hindcast – the results are presented in Fig. 7. They show that when large ($>\pm 2\sigma$) ensemble mean regional rainfall signals are forecast, there are large increases in the probability of observed extreme quantile outcomes and therefore that opportunities exist to develop more confident warnings of seasonal extremes.

Reviewer #2 (Remarks to the Author):

Review of “Windows of opportunity for predicting seasonal climate extremes : Pakistan floods of 2022” by Dunstone et al. submitted to Nature Communications (April 2023)

This paper investigates possible improvements of seasonal forecasts of tropical rainfall 1-4 months mostly from the case study of the extreme JJA season in 2022 in Pakistan. This season was somewhat well predicted but the amplitude of the extreme positive anomaly was underestimated. As seasonal forecasts use ensemble of simulations, the extremes are, by definition, underestimated as soon as the ensemble shows some spread, which is always observed. The authors propose to use supplementary experimental simulations (based on the modification of the initial atmospheric state), when an extreme event is expected as in the boreal spring of 2022, to increase the confidence of the predictions of extremes. It may be an extremely helpful exercise for any impact study related to the occurrence of this extreme. Nevertheless, I do not find here a major and novel result demonstrating a practical improvement of seasonal forecasts. The main issue is that I do not really understand how the increased confidence related to the supplementary simulations is practically implemented. I mean that I do not see how the idealized experiments add something new to our knowledge about air-sea coupling (since we already know from myriad past diagnostic and numerical studies that the atmospheric initial state is not important when the SST forcing is large as it is expected during extreme ENSO events, so what ?) and, more important, how these idealized experiments are practically used to modify quantitatively the expected probabilistic forecasts ? I add that I do not see what you expect to be different between a seasonal forecast showing a large absolute anomaly and an idealized experiment as the one used in this study. I think it is a rather circular exercise. If the ensemble shows a large anomaly, it means a large (and robust) SST forcing upon the atmosphere, right ? The point of the practical implementation of the information gained from the idealized experiments, somewhat a sort of post-processing the seasonal forecasts, is crucial to corroborate what is announced in the title of this ms. I add also that considering a single case study, even if it is the highest observed seasonal anomaly ever observed, is risky. For example, the figure 6 shows that strong positive anomalies are forecasted also over NE India and Bangladesh. Then, the 2022 monsoon was anomalously dry over this area. I am curious if the supplementary simulations show also positive forecasts here. If so, it casts some doubts about the real advantage (or the robustness) of the supplementary experiments to gain confidence in the seasonal forecasts, since in this case, the forecasts for the same season over a close region to Pakistan is totally wrong. It may be different if the authors are able to demonstrate the advantage of supplementary idealized experiments to get a better forecast and explain clearly how these better forecasts are expressed in

practical terms. They can choose to consider the same country with other extreme ENSO years or showing another close test area. So, by now, I found the idea and storyline interesting but I am not convinced by the results, especially how the idealized experiments are practically used to alter the probabilistic forecasts from the seasonal exercise. Another main concern is that the authors consider three possible drivers of the anomalous seasonal rainfall in Pakistan (La Nina event, zonal shift of the west Pacific anticyclone and meridional shift of the subtropical jet) without any clear description and explanation of their relationships, except the fact that both atmospheric indices are independent to each others. Are these atmospheric indices related to the ENSO phenomenon ? SO in summary, I think that this paper needs at least a major revision.

We thank the reviewer for their helpful and detailed comments. The key point we are making is that seasonal forecasts provide more useful information than is currently appreciated, and that forecasts of extremes should in future be taken much more seriously in order to ameliorate their impacts. We argue that there are 'windows of opportunity' when extremes are much more predictable than average, and that understanding the physical drivers will enable more confidence to be built in these situations to provide 'actionable' information to planners. The perturbation experiments are one of several steps we propose for building confidence in the forecasts .

We are not proposing to modify forecast probabilities, but rather that windows of opportunity exist when we should also look beyond tercile (or quintile) probabilities to assess much more extreme outcomes – i.e. that sometimes forecast signals for more extreme outcomes exist (due to predictable large-scale changes in dynamical circulation) which are not currently being exploited (e.g. in current WMO regional seasonal outlooks). Having identified these we argue that more physical understanding of signals should be undertaken so that we can further build confidence in these extreme signals (which may be rare or often unprecedented in the hindcast period) and this information would then feed into part of an expert forecasting judgement.

We agree that further assessment of our proposed approach is needed and have carried out new analysis to assess how well the proposed tool (e.g. Fig. 6) verifies over the 44 year hindcast period (1979-2022) and show this in a new Fig. 7. This shows that when extremes are forecast the outer quintile is actually the most likely outcome, happening in over 50% of cases and an outer decile extreme outcome occurs in over one third of cases, i.e. over three times the 1in10 climatological probability. At the same time, we see a reduction in the neutral outcomes (middle three quintiles) and a large reduction in the opposite extreme quantile outcome. We further show how these benefits increase as a function of the minimum average correlation threshold showing that hindcast skill provides additional information to build confidence.

Whilst this is just an example of one possible approach, we think it nicely demonstrates that such regional examination of predicted extremes using this tool has significant potential to warn of some impending climate extremes by taking advantage of windows of opportunity that are not currently being fully harnessed. We further note that this evaluation is would be followed by the more interactive physical analysis (i.e. step 2/3 of the proposed tool - examining model teleconnections, past performance and possible perturbation experiments) that would act as an additional filter.

More specific concerns

- line 80-82: is there an available reference for that statement ?

Yes, we have added a footnote HTML link to a World Bank press release which is the source of these numbers and leave it to the journal to decide whether the most appropriate format for this (reference or footnote).

- Fig. 1: perhaps, you can state precisely that “summer” is JJA. Even it is stated in the method section, the captions should be more precise about the parameters of the scheme (initialization date, size of the ensemble, period of the retrospective forecasts, etc.) since a figure with its caption should be understood alone.

Thanks we agree, this has been added to the Fig.1 caption.

- line 128: initialization is May 1st ?

Yes, as stated on L132.

- line 188: you stated before that you have a 44-yr ensemble for one model. It is enough to see what happens in other extreme La Nina events.

Yes, but not a sufficient number for robust state dependent statistics/verification.

- Fig. 2d.e: color are for zonal component only ? The caption is not fully clear.

Yes, we’ve amended the caption to make this clear – thanks.

- line 202: from which test ?

Thanks, we’ve added this information into all figure captions.

- line 204: do you have quantitative estimates (RMSE ? Pattern correlations ?) of the „similarity“ ?

We have calculated the pattern correlation between the forecast and observed zonal wind anomalies over the region shown in Fig. 2a,b and find a strong $r=0.7$ correlation.

- Fig. 4: the magenta arrows are difficult to depict.

We tried green but decided these worked better.

- line 274: which indicator for ENSO ? SST indices ? From which initialization ?

Now added to Methods.

- line 311: are these atmospheric drivers related to ENSO ?

- line 358 and 361-362: same comment as line 311.

We use perturbation experiments to assess the impact of tropical Pacific SSTs on atmospheric circulation in panel (c) of Figs. 2,3,4, which shows that the summer 2022 La Nina was a significant driver of both the positive WNPSH and poleward sub-tropical jet anomalies. We have now made this more explicit in L369-378. The average correlations across all years between the Nino3.4 index and the WNPSH and STJshift indices are both similar, giving modest but significant negative correlations ($r=-0.3, p=0.05$), showing that a summer La Nina generally promotes positive WNPSH and a poleward shift of the sub-tropical shift.

- line 317: This score should be cross-validated, but this is perhaps the case ?

We do not cross-validate this as we are not proposing to use this multiple linear regression model in a forecast prediction capacity but instead to gain an understanding of the key dynamical drivers – and for this purpose all previous years would be available.

- line 365: I do not understand how the gain in confidence is implemented in the seasonal forecasts ?

This information regarding our physical understanding of the driving mechanisms of the forecast extreme would need to be interpreted by expert judgement and we’ve added this in L458. We note that even without this step, the new analysis shown in Fig. 7 shows that the proposed tool would still be a useful and skilful approach to identify some possible extreme events.

- line 384: I do not understand if this map has been made from the seasonal forecasts only ? From

the SF + supplementary experiments ?

Just from the seasonal forecast alone, the map simply shows the model (DP3) ensemble mean standardised predicted rainfall anomalies split over the 237 land regions – we’ve now made this explicit in L404.

Reviewer #3 (Remarks to the Author):

In this paper, Dunstone et al. explore the extreme seasonal rainfall associated with the 2022 Pakistan floods as an example of how windows of opportunity can be used to issue more confident forecasts of extreme climate events. The Met Office DePreSys3 near-term prediction system is used to forecast Pakistan rainfall. Eight additional operational seasonal prediction systems are also assessed. Focusing on upper quintile and decile probabilities, the authors show that signals for extreme rainfall were widely present in seasonal prediction systems in summer 2022 and could have been identified had a fuller examination (beyond the upper tercile) of the forecast distribution been considered. Confidence in the DePreSys3 forecasts is established through an understanding of the physical drivers of the Pakistan rainfall, developed using perturbation experiments to isolate the effects of La Niña, and a multiple linear regression model relating rainfall to low- and upper-level circulation. Based on their results, the authors suggest that near-term climate forecasts should be routinely monitored for extreme signals and propose an interactive tool to aid with this.

The paper is well-written with the methods and results clearly presented. Given the extent of the damage that can be caused by extreme events (as discussed in the Introduction), issuing trustworthy and actionable warnings is of the utmost importance. There is great potential in the methods discussed here to develop confident forecasts of climate extremes. Thus, I believe this paper represents a valuable contribution to the field and will be of wide interest, meriting publication in Nature Communications.

Thank you for your review.

I have only a small number of very minor comments/corrections:

The authors might consider referring to DePreSys3 using an abbreviation such as “DP3” (as in Ref. 18) instead of the “model” to distinguish it more clearly from the other seasonal prediction systems and the empirical model also used in the study.

Agreed, this would indeed make things clearer and so we have adopted this terminology, thanks for the suggestion.

Line 80: Correct “devasting” to “devastating.”

Thanks, corrected.

Line 146: Change “blue curve” to “blue line” for consistency with the next paragraph.

Changed.

Line 153–4: “While we note that any shift will give a monotonically increasing factor on the likelihood to higher and higher percentiles [...]” I am not sure I fully understand what is meant by this. Could you please explain?

This has now been removed.

Line 281: Remove the citation in parentheses.

Removed.

REVIEWERS' COMMENTS

Reviewer #1 (Remarks to the Author):

Review comments of the revised manuscript NCOMMS-23-02675A entitled “Windows of opportunity for predicting seasonal climate extremes: Pakistan floods of 2022” by Dunstone et al. submitted to nature communications

I have gone through the revised manuscript and found that the authors have given satisfactory explanations to the suggested comments. I propose to accept the manuscript after incorporating a few minor comments, which are appended below.

General comments

1. Give highlights of results in the abstract, now it is lacking
2. I still believe the authors need to give a brief about the prediction of seasonal extremes as they stated in Fig. 6 caption “DP3 ensemble mean standardised predicted rainfall anomalies for summer 2022”
3. Need to revise the maps of Figures 6, S2 and S3, in which the political map of India is not correct.

Reviewer #2 (Remarks to the Author):

Review of R1 of « Windows of opportunity for predicting seasonal climate extremes: Pakistan floods of 2022 » by Dunstone et al. (2023)

The authors replied convincingly to most of the concerns. I have only minor comments on this revised version.

Line 189 : perhaps just remind here what is the correlations between Pakistan monsoon rainfall (PMR) and ENSO indices in April or May ? You may also perhaps compute the correlation between observed PMR and D3 outputs from May initialisation when April ENSO indices are, for example, outside the [-0.5-0.5] range ?

Line 193 : I agree in theory but you can also use the 40-member ensemble to give a score about the potential predictability of, for example, La Nina year. I mean, that if the predictability is higher during la Nina year (such hypothesis makes sense) then you should achieve a higher signal-to-noise ratio in the 40-member ensemble.

Line 211-213 : the term « over this region » on line 213 is here ambiguous since I do not think that 850 hPa climatological winds are westerly in JJA « from the Bay of Bengal, across northern Indian, towards Pakistan »

Line 246 : « reversed the climatological westerly flow » do you mean that the raw 250 hPa winds were easterly in JJA 2022 ?

On figure 4, the vertical velocity is not very clear especially when arrows are superimposed on dark blue colors. I suggest to show the climatological winds in SI and shows here the vertical velocity as shadings perhaps superimposed on the climatological values as contours ?

Line 281-282 : « of summer ENSO variability » : do you mean a specific ENSO indices as Nino 3.4 SST index ?

Line 290-292 : « We suggest that such perturbation experiments to establish physical drivers and mechanisms could in theory be performed in real-time in order to be able to issue more confident warnings of impending extreme events. » : I agree but which objective criteria can be used to launch such perturbation experiments ? The PDF of the simulated anomalies ? A specific combination of drivers ? I think that this is a key-point of your paper and that you need to discuss it a little bit more. This comment applies also to what is written on line 311 « these experiments had been run in real-time » : OK, but which objective criteria will be used to launch these experiments ? Idem on line 383 or line 425 (« when appropriate » is too vague).

Line 294-295 : the choice of 2021 as « near-neutral » seems not optimal since the boreal spring was involved into the 3-year Nina event ? 2020 isn't better ?

Line 410-412 : « The standardised forecast rainfall anomalies for summer 2022 show that Pakistan is one of the only four regions with a $>\pm 3\sigma$ signal, whilst a further 21 regions have $>\pm 2\sigma$ rainfall anomalies » : at the end, what was the observed rainfall anomalies for these 24 regions shown on

Fig. 6 ? I think that (at least) Bangladesh, and probably NE India, was anomalously dry for these seasons while DP3 predicts anomalously wet conditions. I do not think also that South Sudan was so anomalously wet. The map on <https://iridl.ldeo.columbia.edu/maproom/Global/Precipitation/Seasonal.html?T=June-Aug 2022> shows near zero anomalies. I think that any forecast model should not hide failures and it suggests that even large predicted extreme could be wrong (and not very far from Pakistan for another area involved in the same regional system).

We thank the Editor and the Reviewers for their further consideration of our manuscript and are pleased that we have been able to satisfactorily respond to the helpful comments and suggestions. We have addressed as many of the remaining comments as possible below.

Reviewer #1 (Remarks to the Author):

Review comments of the revised manuscript NCOMMS-23-02675A entitled “Windows of opportunity for predicting seasonal climate extremes: Pakistan floods of 2022” by Dunstone et al. submitted to nature communications

I have gone through the revised manuscript and found that the authors have given satisfactory explanations to the suggested comments. I propose to accept the manuscript after incorporating a few minor comments, which are appended below.

General comments

1. Give highlights of results in the abstract, now it is lacking

We have amended the abstract very slightly, in response to this and editorial suggestion – however, overall we are happy that it accurately reflects and summarises the paper.

2. I still believe the authors need to give a brief about the prediction of seasonal extremes as they stated in Fig. 6 caption “DP3 ensemble mean standardised predicted rainfall anomalies for summer 2022”

We have added some extra descriptive text to Fig. 6 caption and to Methods.

3. Need to revise the maps of Figures 6, S2 and S3, in which the political map of India is not correct. The regions used are those defined by Stone et al 2019, via their shapefiles, and are designed to be approximately equal area and along administrative boundaries. However, we have added this sentence to Methods “...we note that the authors remain neutral with regard jurisdictional claims in all maps” and we expect the journal will add a similar statement as they did in another recent article that also used these regions (Thompson et al 2023, Nature Commun, <https://www.nature.com/articles/s41467-023-37554-1>): “Publisher’s note Springer Nature remains neutral with regard to jurisdictional claims in published maps and institutional affiliations.”

Reviewer #2 (Remarks to the Author):

Review of R1 of « Windows of opportunity for predicting seasonal climate extremes: Pakistan floods of 2022 » by Dunstone et al. (2023)

The authors replied convincingly to most of the concerns. I have only minor comments on this revised version.

Line 189 : perhaps just remind here what is the correlations between Pakistan monsoon rainfall (PMR) and ENSO indices in April or May ? You may also perhaps compute the correlation between observed PMR and D3 outputs from May initialisation when April ENSO indices are, for example, outside the [-0.5-0.5] range ?

This is strongly dependent on whether you include 2022 or not (as we already describe at the start of the section for Pakistan rainfall skill). We think that adding this information would disrupt the flow of the manuscript and not add much for the reader. The main point we're making here is that we do not have enough samples of strong summer La Nina events to quantify Pakistan rainfall skill during such events. We have added 'strong' before La Nina to make this point clearer.

Line 193 : I agree in theory but you can also use the 40-member ensemble to give a score about the potential predictability of, for example, La Nina year. I mean, that if the predictability is higher during la Nina year (such hypothesis makes sense) then you should achieve a higher signal-to-noise ratio in the 40-member ensemble.

We agree in principle if the model was 'perfect'. However, many studies have found perfect predictability measures to underestimate the skill that could potentially be achieved in predicting the real world (e.g. Eade et al 2013), due to the often spuriously weak dynamical signals/teleconnections in current climate models. There is also a lack of independence to be concerned with, i.e. members would have common variability in other parts of the climate system.

Line 211-213 : the term « over this region » on line 213 is here ambiguous since I do not think that 850 hPa climatological winds are westerly in JJA « from the Bay of Bengal, across northern Indian, towards Pakistan »

ERA5 climatology shows that they are mostly westerly here, with only a relatively small region of climatological easterlies in the far north, but we have now amended this sentence accordingly to avoid possible confusion.

Line 246 : « reversed the climatological westerly flow » do you mean that the raw 250 hPa winds were easterly in JJA 2022 ?

Yes, but to be more accurate this reversal is only over the southern Tibetan Plateau as discussed in He et al 2023. We have added 'southern' to make this clearer that it is not the entire Tibetan Plateau.

On figure 4, the vertical velocity is not very clear especially when arrows are superimposed on dark blue colors. I suggest to show the climatological winds in SI and shows here the vertical velocity as shadings perhaps superimposed on the climatological values as contours ?

We have considered this but think our implementation is still the best compromise.

Line 281-282 : « of summer ENSO variability » : do you mean a specific ENSO indices as Nino 3.4 SST index ?

Yes, we have now added "(Niño3.4 region skill: $r=0.86$, $p<0.001$)".

Line 290-292 : « We suggest that such perturbation experiments to establish physical drivers and mechanisms could in theory be performed in real-time in order to be able to issue more confident warnings of impending extreme events. » : I agree but which objective criteria can be used to launch such perturbation experiments ? The PDF of the simulated anomalies ? A specific combination of drivers ? I think that this is a key-point of your paper and that you need to discuss it a little bit more. This comment applies also to what is written on line 311 « these experiments had been run in real-time » : OK, but which objective criteria will be used to launch these experiments ? Idem on line 383 or line 425 (« when appropriate » is too vague).

We have added (L284) that this would be when large-scale SST patterns show significant anomalies and hence a predictable influence on atmospheric circulation may occur.

Line 294-295 : the choice of 2021 as « near-neutral » seems not optimal since the boreal spring was involved into the 3-year Nina event ? 2020 isn't better ?

Summer 2020 saw a stronger summer La Nina event than summer 2021 and so would have been a worse choice.

Line 410-412 : « The standardised forecast rainfall anomalies for summer 2022 show that Pakistan is one of the only four regions with a $>\pm 3\sigma$ signal, whilst a further 21 regions have $>\pm 2\sigma$ rainfall anomalies » : at the end, what was the observed rainfall anomalies for these 24 regions shown on Fig. 6 ? I think that (at least) Bangladesh, and probably NE India, was anomalously dry for these seasons while DP3 predicts anomalously wet conditions. I do not think also that South Sudan was so anomalously wet. The map on

<https://iridl.ldeo.columbia.edu/maproom/Global/Precipitation/Seasonal.html?T=June-Aug> 2022 shows near zero anomalies. I think that any forecast model should not hide failures and it suggests that even large predicted extreme could be wrong (and not very far from Pakistan for another area involved in the same regional system).

We agree that an evaluation of the summer 2022 forecast using this tool would be useful and so we have added an additional paragraph on this:

“In summer 2022 there are 35 regions (Fig. 6) with a predicted $>\pm 2\sigma$ rainfall anomaly, of which 16 show evidence for significant skill in the hindcast period ($r > 0.25$, see Fig. S3). Examining the observed GPCP summer 2022 outcomes, we find that 12 of the 16 (75%) regions experienced a correct sign outer quintile rainfall event, of which eight (50% of the 16) were outer decile and four (25% of the 16) were outer 5% rainfall extremes (including Pakistan). We note that the frequency of these extreme observed quantile outcomes are considerably higher than climatological expectation, in agreement with the analysis in Fig. 7, and that the four of 16 that did not experience an observed outer quintile event were near-neutral outcomes (fell into the middle three quintiles).”

We note that, as shown in Fig. S3, NE India and Bangladesh do not show prior evidence of significant DP3 rainfall skill and so the wet 2022 forecast signals over these regions would not have been given as much weight as those for Pakistan which does show significant hindcast skill (even before 2022).